# Spatial Analysis of Schistosomiasis in Hunan and Jiangxi Provinces in the People’s Republic of China

**DOI:** 10.3390/diseases10040093

**Published:** 2022-10-19

**Authors:** Kefyalew Addis Alene, Catherine A. Gordon, Archie C. A. Clements, Gail M. Williams, Darren J. Gray, Xiao-Nong Zhou, Yuesheng Li, Jürg Utzinger, Johanna Kurscheid, Simon Forsyth, Jie Zhou, Zhaojun Li, Guangpin Li, Dandan Lin, Zhihong Lou, Shengming Li, Jun Ge, Jing Xu, Xinling Yu, Fei Hu, Shuying Xie, Donald P. McManus

**Affiliations:** 1Faculty of Health Sciences, Curtin University, Perth 6102, Australia; 2Infection and Inflammation Program, QIMR Berghofer Medical Research Institute, Brisbane 4006, Australia; 3School of Population Health, University of Queensland, Brisbane 4072, Australia; 4Department of Global Health, Australian National University, Canberra 0200, Australia; 5National Institute of Parasitic Diseases, Chinese Center for Disease Control and Prevention, NHC Key Laboratory of Parasite and Vector Biology, WHO Collaborating Centre for Tropical Diseases, National Center for International Research on Tropical Diseases, Shanghai 200025, China; 6Hunan Institute of Schistosomiasis Control, Yueyang 414000, China; 7Swiss Tropical and Public Health Institute, CH-4051 Allschwil, Switzerland; 8University of Basel, CH-4003 Basel, Switzerland; 9Jiangxi Institute of Parasitic Diseases, Nanchang 330096, China

**Keywords:** People’s Republic of China, *Schistosoma japonica*, schistosomiasis, spatial analysis

## Abstract

Understanding the spatial distribution of schistosome infection is critical for tailoring preventive measures to control and eliminate schistosomiasis. This study used spatial analysis to determine risk factors that may impact *Schistosoma japonicum* infection and predict risk in Hunan and Jiangxi Provinces in the People’s Republic of China. The study employed survey data collected in Hunan and Jiangxi in 2016. Independent variable data were obtained from publicly available sources. Bayesian-based geostatistics was used to build models with covariate fixed effects and spatial random effects to identify factors associated with the spatial distribution of infection. Prevalence of schistosomiasis was higher in Hunan (12.8%) than Jiangxi (2.6%). Spatial distribution of schistosomiasis varied at pixel level (0.1 × 0.1 km), and was significantly associated with distance to nearest waterbody (km, β = −1.158; 95% credible interval [CrI]: −2.104, −0.116) in Hunan and temperature (°C, β = −4.359; 95% CrI: −9.641, −0.055) in Jiangxi. The spatial distribution of schistosomiasis in Hunan and Jiangxi varied substantially and was significantly associated with distance to nearest waterbody. Prevalence of schistosomiasis decreased with increasing distance to nearest waterbody in Hunan, indicating that schistosomiasis control should target individuals in close proximity to open water sources as they are at highest risk of infection.

## 1. Introduction

Schistosomiasis, caused by an infection with blood flukes of the genus *Schistosoma*, is a neglected parasitic disease that leads to a considerable level of morbidity [1]. It has been estimated that schistosomiasis afflicts more than 240 million people globally, with an estimated loss of 1.4 million disability-adjusted life years (DALYs) in 2017 [2]. The highest prevalence of the disease has been reported in tropical and subtropical regions, including in sub-Saharan Africa, the Middle East, Southeast Asia, South America, and the Caribbean regions [3]. In the People’s Republic of China and the Philippines, schistosomiasis is caused by *Schistosoma japonicum*. Humans acquire *Schistosoma* infection when they come into contact with freshwater bodies that contain the infectious stage of the parasite [4]. People living in areas with poor sanitation and a lack of a clean drinking water supply are at increased risk of contracting schistosomiasis.

The transmission of schistosomiasis is complex and influenced by individual, household, and ecological factors [5]. Individual-level demographic, behavioural, and socioeconomic risk factors have been well described in the literature. For example, in a recent study carried out in the People’s Republic of China, it was shown that low wealth index was associated with schistosomiasis [6]. Other previous studies have also identified individual-level factors that are closely associated with schistosome infections, such as poor sanitation, lack of drinking water, poor socioeconomic conditions, and demographic factors [7,8]. However, identifying ecological-level factors associated with schistosome infection is also important when implementing public health interventions based on the local context. Previous research has indicated that ecological-level environmental and climatic factors such as temperature influence the spatial distribution of schistosomiasis [9,10,11,12]. Being a water-based disease, distance to a waterbody might also influence the spatial distribution and transmission of schistosomiasis [13,14].

There is a range of interventions currently available for the prevention, control, and elimination of schistosomiasis in endemic countries. These interventions include preventive chemotherapy with praziquantel, improvements in clean water supply and sanitation, snail and definitive host control, health education and health promotion, and specific targeting of high-risk populations [15,16,17]. The global strategy for schistosomiasis control focuses on reducing disease through periodic and targeted treatment with praziquantel of affected populations [15]. Understanding the spatial distribution of schistosomiasis is important to implement these intervention strategies. Risk maps showing the geographical distribution of schistosomiasis can inform decisions of policymakers on the rational allocation of resources for targeted interventions. Risk mapping is especially important for schistosomiasis control programmes in the People’s Republic of China and elsewhere, as the strategy is shifting from morbidity control to elimination, thus breaking transmission [18].

The aim of this study was to investigate the spatial distribution of schistosomiasis japonica and quantify the relationship with environmental and climatic factors in Hunan and Jiangxi, two of the seven provinces of the People’s Republic of China where the disease remains endemic [6]. Of note, the People’s Republic of China is seeking to eliminate schistosomiasis by 2030, a date revised several times from 2020 to 2025. Despite great strides towards this goal, the 2030 schistosomiasis elimination target set by the central government may yet prove challenging. Thus, this paper is presented to help understand spatial and transmission dynamics in these two endemic regions.

We dedicate this paper to Professor Marcel Tanner in celebration of his 70th birthday. Our piece complements another paper that pertains to short-, mid-, and long-term effects stemming from the 10-year World Bank Loan Project for schistosomiasis control in the People’s Republic of China [19]. Together, these two papers honour Tanner’s research-cum-action work at the Swiss Tropical and Public Health Institute, the University of Basel and various other institutions to improve people’s health and wellbeing in different parts of the world.

## 2. Materials and Methods

### 2.1. Study Design and Settings

This study was conducted using an ecological study design where associations between dependent and independent variables were measured at area levels. The study used extensive survey data collected in Hunan and Jiangxi Provinces. A detailed description of the survey is available elsewhere [6]. Briefly, the baseline survey for a longitudinal cohort study was carried out across 16 villages around the Dongting and Poyang lakes in Hunan and Jiangxi Provinces in 2016 (Figure 1). Survey participants were randomly selected from each village. Stool samples were collected from the survey participants. Total DNA was extracted from human stool samples, using QIAamp DNA mini kits (Qiagen; Hilden, Germany) and real-time polymerase chain reaction (qPCR) assays performed [6]. Data were collected from 3924 study participants aged between 5 and 70 years. Demographic data were collected with a pre-tested questionnaire when the participants were enrolled in the study [6]. Global positioning system (GPS) data of study households were captured during the survey using Garmin Handheld GPS (Olathe, KS, USA). Data for water sources were also obtained from the Global Lakes and Wetlands Database [20].

### 2.2. Ethics

Written informed consent was received from all individuals enrolled in the study, and from parents/guardians of minors. Ethical approval for human and animal work was provided by the Ethics Committees of QIMR Berghofer Medical Research Institute (Human Research Ethics Committee, reference number P524; Animal Ethics Committee, reference number A1003-601), Hunan Institute of Parasitic Diseases (HIPD), Jiangxi Institute of Parasitic Diseases (JIPD), and the National Institute of Parasitic Diseases (NIPD), Shanghai.

### 2.3. Data Sources

We used different data sources for the primary outcome and exposure variables. Our primary outcome measure was prevalence of schistosomiasis, as determined by qPCR. These data were obtained from the human schistosomiasis survey [6]. GPS data showing the locations of the survey participants were converted to a decimal degree format.

Our exposures of interests were (i) distance to nearest waterbody; (ii) climatic variables; and (iii) access to healthcare. These variables were selected based on evidence of association with schistosome infection from previous studies and based on the availability of province-wide representative data [3,4,21].

The geodesic distance between each survey respondent’s home and the closest waterbodies (i.e., lakes, ponds, rivers, and streams) was calculated using ArcGIS version 10.6.1 software (ESRI; Redlands, CA, USA). Different to the Euclidean distance, the geodesic distance captures the curvature of the Earth [22]. The average distance between natural village and the nearest waterbody was calculated for the geospatial analysis. Of note, ‘natural’ villages form identifiable clusters within each administrative village—they share a homogeneous ecology and lifestyle. There are 6–16 natural villages per administrative village. First, we calculated a centroid for the natural village by taking the average values of the GPS coordinates of each household from that natural village participating in the survey. Second, a buffer zone was created for the natural village, with varying radii (i.e., 100 m, 200 m, 500 m, and 1 km; Appendix A). We then created a small rectangular polygon (fishnet) covering the whole territory of Jiangxi and Hunan Provinces at a spatial resolution of 0.1 × 0.1 km^2^ using tools in ArcGIS. The distance to the nearest waterbody was calculated at the centroid of each natural village with 500 m radii. Mean values of distance to the nearest waterbody were extracted for the cell centroids contained in each buffer zone. For subsequent spatial prediction, buffers were generated for the centroid of each grid cell across the two study provinces (using the same radii as for the observed data) and mean distance to the nearest waterbody was extracted within each buffer to create a variable for spatial prediction.

Climatic variables such as mean temperature, precipitation, and solar radiation were obtained from the WorldClim database [23]. Data on altitude were obtained from the Shuttle Radar Topography Mission (SRTM) [24] and a shapefile containing waterbodies in Hunan and Jiangxi Provinces was obtained from DIVA-GIS [25]. In addition, data on healthcare access (i.e., walking travel times in min to the nearest general practitioner health facilities) were obtained from the Malaria Atlas Project (MAP) [26]. All data were extracted at a spatial resolution of 1 × 1 km. A polygon shapefile for the administrative boundaries of Hunan and Jiangxi Provinces was obtained from Global Administrative Areas (GADM) [27]. The data obtained from different sources were collated in a geographical information system, ArcGIS version 10.6.1 (ESRI Inc.; Redlands, CA, USA) (Figure 2). The data sources of the covariates with their definitions are provided in the Appendix A.

### 2.4. Statistical Analysis

We undertook descriptive analyses, calculating means with standard deviations (SD) for normally distributed continuous variables, medians with an interquartile range (IQR) for non-normally distributed continuous variables, and percentages (%) for categorical variables. Since the independent variables have different units and scales of measurement with unknown threshold effects, they were standardized to a Z-scale based on their mean and SD. This method also helped with identifiability in the estimation of the posterior distribution of the coefficients. All independent variables were tested for multi-collinearity, and those variables with a variance inflation factor (VIF) > 7 were excluded from the final model.

A nonspatial logistic regression model was first fitted between independent variables and schistosomiasis prevalence. Variables with a *p*-value < 0.02 in the nonspatial regression model were included in the final geospatial model. A geospatial model was fitted for schistosomiasis prevalence survey data using both fixed covariate effects and random spatial effects based on the approach of model-based geostatistics [28].

### 2.5. Geospatial Analysis

Two models were constructed separately for Hunan and Jiangxi Provinces. The proportion of *S. japonicum* infection at each surveyed location *j* as the outcome variable was assumed to follow a binomial distribution:$$\;Y_{j}\sim \;Binomial\;\left(n_{j},p_{j}\right)\;$$
where $Y_{j}$ is the number of schistosomiasis positive, $n_{j}$ is the number of individuals screened for schistosomiasis, and $p_{j}$ is the predicted schistosomiasis prevalence at location j. Mean predicted schistosomiasis prevalence was modelled via a logit link function to a linear predictor as follows:$$logit\left(p_{j}\right)=\alpha +\overset{z}{\underset{z=1}{\sum }}\beta_{z}X_{z,j}+{\;}_{\;}\zeta_{j}$$
where α is the intercept, *β* is a matrix of covariate coefficients, X is a design matrix of z covariates, and $\zeta_{j\;}$ are spatial random effects modelled using a zero-mean Gaussian Markov random field (GMRF) with a Matérn covariance function. The covariance function was defined by two parameters: the range ρ, which represents the distance beyond which correlation becomes negligible, and σ, which is the marginal SD [29,30]. Non-informative priors were used for α (uniform prior with bounds –∞ and ∞) and we set normal priors with mean = 0 and precision (the inverse of variance) = 1 × 10^−4^ for each *β*. We used default priors for the parameters of the spatial random field [31]. Parameter estimation was done using the Integrated Nested Laplace Approximation (INLA) approach in R (R-INLA) [29,30]. Sufficient values (i.e., 150,000 samples) from each simulation run for the variables of interest were stored to ensure full characterization of the posterior distributions. The Watanabe-Akaike information criterion (WAIC) statistic was used to select the best-fitting model (Appendix A). The geospatial analysis and the descriptive analysis were conducted using R statistical software version 4.0.2 (Boston, MA, USA).

### 2.6. Spatial Prediction

Predictions of the prevalence of schistosomiasis at unsampled locations was made at a small geographic scale (approximately 0.1 km^2^) by interpolating the geostatistical random effects and adding these to the sum of the products of the values of the fixed effects at each prediction location [32]. These provide prediction surfaces that show the prevalence of schistosomiasis for all prediction locations in Hunan and Jiangxi Provinces.

## 3. Results

A total of 2349 households with unique GPS coordinates in 16 administrative villages were included in this study [6]. Of these, 1099 (46.7%) were from Hunan Province and the remaining 1250 (53.2%) from Jiangxi Province. The mean age of the study participants was 47.2 years (SD = 12.3 years). The geographic locations of the administrative villages are presented in Figure 1. The median distance of the household to the nearest waterbody was 0.57 km (interquartile range (IQR) = 0.20–1.29 km). The mean walking travel times to the nearest health facilities, a proxy measure of healthcare access, was 1.8 h (SD = 1.8 h) for Hunan Province and 2.4 h (1.8 h) for Jiangxi Province. Table 1 summarises the characteristics of variables included in the study.

### 3.1. Prevalence of Schistosomiasis at the Province and Village Levels

Full prevalence and intensity of infection data for the 16 administrative villages in the two provinces can be found elsewhere [6]. Table 2 shows the observed prevalence of schistosomiasis at province and village levels in Hunan and Jiangxi Provinces. Of the total of 2349 study participants, 174 were confirmed positive for *S. japonicum* infection by qPCR. The overall schistosomiasis prevalence was 7.4% and was higher in Hunan Province (12.8%) compared to Jiangxi Province (2.6%) [6]. Considerable variation was also observed in the prevalence of schistosomiasis at the village level, with the highest prevalence observed in Xiangjiang (39.3%) and Longwang (27.9%) villages and, apart from Dingshan where no infected individuals were identified, the lowest recorded prevalence was in Huanggin (1.2%) village [6].

### 3.2. Spatial Distribution of Schistosomiasis

There was substantial variation in the predicted prevalence of schistosomiasis at the pixel level in both provinces (Figure 2). The mean predictive prevalence at pixel level ranged from 4% to 67% in Hunan Province and from 0.2% to 79% in Jiangxi Province. It should be noted that we did not have data points in areas outside the survey boundary and, therefore, estimates made for such areas are more speculative and rely on data from within the survey boundaries and their relationship with covariates. The SD for the predicted prevalence was high outside the survey locations, indicating a high level of uncertainty in these areas (Appendix A).

### 3.3. Ecological-Level Factors Associated with the Spatial Distribution of Schistosomiasis

Table 3 shows the association between ecological level factors and schistosomiasis prevalence in Hunan and Jiangxi Provinces, obtained from nonspatial multivariate logistic regression models. The models show that schistosomiasis prevalence was negatively associated with distance to waterbody, distance to the nearest health facility (in min), and altitude (in m) in Hunan Province (Table 3). In Jiangxi Province, distance to waterbody was negatively associated with schistosomiasis prevalence; whereas temperature was positively associated with schistosomiasis prevalence (Table 3).

In the Bayesian spatial logistic regression model, after incorporating the spatial random effect in the geostatistics model, distance to neared waterbody was negatively associated with schistosomiasis prevalence in Hunan Province (Table 4). This means the prevalence of schistosomiasis decreased with increasing distance to the nearest waterbody in this province (Figure 3). However, it was not significant in Jiangxi Province. In Jiangxi Province, temperature was associated with schistosomiasis prevalence (Table 4). Other ecological level factors included in the models such as distance to health facilities and altitude were not significantly associated with schistosomiasis prevalence in either province (Table 4). Figure 2 presents the predicted prevalence maps of *S. japonicum* infections in Hunan and Jiangxi Provinces as calculated by spatial risk modelling. The maps showed that the spatial distribution of schistosomiasis substantially varied at small geographical scale. The WAIC value corresponding to different model specifications is available in the Appendix A.

Maps of individual variables included in our models such as nearest distance from waterbody, altitude, which is higher on the borders of each province, and temperature which is high over the majority of both provinces, are presented as Appendix A.

## 4. Discussion

The prevalence of schistosomiasis has dropped significantly in the People’s Republic of China since it was first reported there in 1949, with 12 million people estimated to be infected. More recent estimates indicated 184,943 people were infected in 2013 in all endemic provinces [33,34,35]. In 2018 there were an estimated 24,986 human cases in Hunan [36,37], while only 429 positives were identified in Jiangxi Province from 10,700 individuals tested [38]. As the People’s Republic of China has moved towards schistosomiasis elimination, spatially explicit risk maps will be essential in order to pinpoint transmission areas where disease is likely to occur, thereby allowing for targeted control and elimination efforts.

In general, we found that distance from household to the nearest waterbody was negatively associated with *S. japonicum* infection in Hunan Province, i.e., the closer a household was to open freshwater sources, the higher the schistosomiasis prevalence (Figure 3). This negative association was significant in Hunan, but not in Jiangxi, although the trend was towards a negative association. This negative association is understandable considering the lifecycle and transmission of *S. japonicum*. Infected *Oncomelania* snails releasing infective cercariae are found in open water sources, and the closer a household is to an open water source, the more contact the householders would have with the water source with increased risk in becoming infected.

An earlier disease mapping project in Hunan Province also reported a significant association between distance to the Dongting Lake and becoming infected with *S. japonicum* [39], although this association was not significant when spatial correlation was taken into account. This previous study used prevalence data based on the Kato-Katz diagnostic method, which is insensitive, and an indirect enzyme linked immunosorbent assay (ELISA) for the presence of anti-soluble egg antigen (SEA) IgG antibodies, which is more sensitive but cannot discriminate between current and past infections. Accordingly, the use of less sensitive and specific diagnostics will likely have influenced the results. Sensitive diagnostics are critical for effective disease control and elimination, and hence, for accurate risk mapping. Whereas the PCR-based assay used in the current study is highly sensitive and specific, it is expensive and thus unlikely to be used routinely as a diagnostic test in schistosomiasis control and elimination programmes. However, as molecular diagnostic methods continue to improve and reduce in price, integration of these methods in routine control will become increasingly feasible.

Access to potable water and other health infrastructure may also have an influence on whether distance to water is a risk factor for schistosomiasis japonica. In the current study, most households had flushable toilets (78.4%), while just over half stated they had access to tap water (54.8%) [6]. Of those who did not have tap water at home, water was accessed via a hand pump (15.8%) or a well (27.8%). Having access to water sources other than natural or open water sources, where the intermediate oncomelanid snail hosts live, will reduce household use of those water sources and thus lessen the risk of infection. Thus, close proximity to water source may be a significant risk factor where water is not accessible in the household, and when other alternatives such as wells are not available, and where the closest natural water source is primarily used for washing and drinking. A previous study in Côte d’Ivoire, West Africa, found that living in close proximity to water sources was significantly associated with *S. mansoni* infection [13]. In that study area, water was supplied via a dam, which was treated before being distributed to households through a network of channels, with most households relying on either well water (63.4%) or tap water (34.4%). However, those living in close proximity to open water sources reported they used the open water sources for irrigation, washing dishes, and laundry. Households that lived further from open sources of water were in neighbourhoods that had better access to tap water and sanitation [13]. In this current Chinese study, most reported water contact reflected farming and/or fishing activities, and few villagers reported using open water sources for washing clothes (4%); although, 18% of respondents reported using open water sources for washing hands/wading [6].

Overall, water contact was high (88%) with most study individuals reporting water contact in the previous 12 months. The most common type of water contact was linked to farming/cutting grass and this was significantly associated with *S. japonicum* infection [6]. The majority (85.5%) of the study population was engaged in farming and/or fishing.

Temperature was the only co-variate found to be positively correlated with *S. japonicum* infection, and only in Jiangxi Province (Table 3). The average temperature between the two provinces is very similar, whereas the prevalence of schistosomiasis in Jiangxi was significantly lower than in Hunan Province. Climate is a major driver for geographical spread of schistosomiasis, particularly in relation to the intermediate snail hosts. In the People’s Republic of China, there are two transmission seasons which are generally associated with annual flooding of the Yangtze River—April to July during peak flooding, and in September to November once flooding begins to recede [40,41]. The rainy season in the People’s Republic of China generally starts in May and ends around September, and strong rains contribute to flooding events. This contrasts with the Philippines (the other major county where *S. japonicum* is endemic), where transmission is year-round due to the absence of a dry season in most endemic areas [42]. It is notable that the average rainfall in Jiangxi is higher than in Hunan (Table 1), although this was not associated in any way with prevalence of schistosomiasis in the study villages. This may be an issue with the small scale of the current study. Rainfall and temperature may influence prevalence on the large scale, but may not be observable in the small scale of this study.

Household survey data, presented in Gordon et al. (2020) [6], showed that the surveyed populations in both provinces were fairly homogenous with similar levels of education, the majority employed as farmers and/or fishermen, and similar socioeconomic levels as determined by a wealth index. In that publication prevalence was higher in households with a lower wealth index [6]. Transmission dynamics have also previously been noted to be similar between the two provinces, at least around the Dongting and Poyang lakes which were surveyed in this study [36].

Our study exhibits some limitations. The survey conducted was not designed for the specific purpose of spatial analysis, and we observed high uncertainty in the predictions where data were spatially sparse, in particular away from the rivers. The spatial predictions in these areas are likely to be less reliable. Therefore, conducting population surveys in these areas, as well as in other parts of the counties in the province where data are spatially sparse, would help assess the spatial variability of the schistosomiasis prevalence and the extent of areas at risk in Hunan and Jiangxi Provinces. Moreover, it was not possible to obtain survey data collected in recent years. However, as the changes in schistosomiasis prevalence are gradual and no large-scale intervention occurred after 2016, we believe that this prevalence data can be used as indicative of the current burden.

## 5. Conclusions

In this study, the spatial distribution of schistosomiasis in Hunan and Jiangxi varied substantially at the provincial, village, and local levels, and was significantly associated with distance to nearest waterbody and climatic factors. We produced maps showing the spatial distribution of schistosomiasis in Hunan and Jiangxi Provinces. These maps may help the local governments to design strategies and target interventions for their schistosomiasis control and elimination programmes. Interventions can be targeted to individuals who live in close proximity to open water sources as they are at greater risk of infection and the negative implications of schistosomiasis japonica. However, further nationwide studies using genomic and survey data might be required to better understand why distance to waterbody was negatively associated with distribution of schistosomiasis in Hunan province but not in Jiangxi Province.

## Figures and Tables

**Figure 1 diseases-10-00093-f001:**
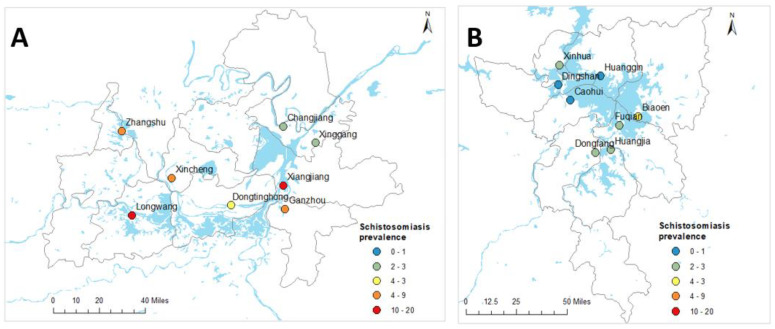
Geographical locations of village centroids and observed prevalence of schistosomiasis in Hunan (**A**) and Jiangxi (**B**) provinces, People’s Republic of China.

**Figure 2 diseases-10-00093-f002:**
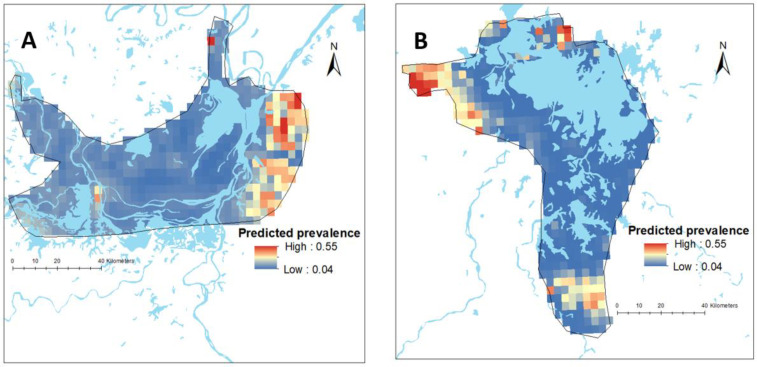
Map showing the predicted spatial prevalence of *Schistosoma japonicum* infections in Hunan (**A**) and Jiangxi (**B**) provinces in the People’s Republic of China (approximately 0.1 km^2^), as calculated by spatial risk modelling.

**Figure 3 diseases-10-00093-f003:**
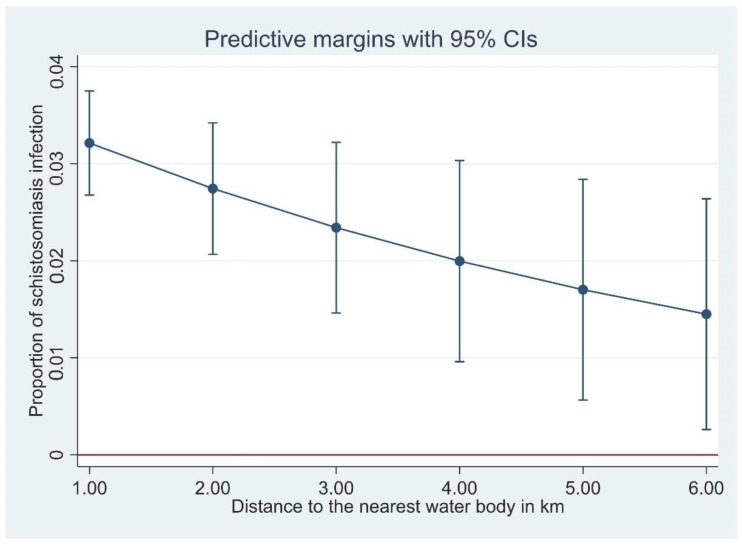
The proportion of *Schistosoma japonicum* infection with distance of household to the nearest waterbody, derived from a logistic regression model.

**Table 1 diseases-10-00093-t001:** Characteristics of the variables included in the study in Hunan and Jiangxi Provinces, People’s Republic of China.

Variable	Hunan Province	Jiangxi Province	Total
Total number of households with GPS coordinates	1099	1250	2349
Number of administrative villages	8	8	16
Number of household members	4367	5657	10,024
Number with *Schistosoma japonicum* infection	141	33	174
Number of male participants	989	1061	2050
Number of female participants	916	958	1874
Age of the survey participants (year), mean (SD)	46.4 (10.5)	47.8 (6.9)	47.2 (12.3)
Distance from household to nearest waterbody (km), mean (SD)	1.22 (1.55)	0.73 (0.94)	0.96 (1.28)
Walking distance to health facilities (min), mean (SD)	108 (108)	143 (107)	126 (109)
Average monthly temperature (°C), mean (SD)	17.2 (0.1)	17.6 (0.1)	17.4 (0.2)
Average monthly precipitation (mm), mean (SD)	107.6 (3.1)	129.4 (3.7)	119.2 (11.4)
Average monthly solar radiation (kJ m^−2^ day^−1^), mean (SD)	13,629 (87)	14,703 (93)	14,201 (544)
Elevation (m), mean (SD)	34.3 (5.7)	19.8 (17.5)	26.6 (15.2)

GPS, global positional system; SD, standard deviation.

**Table 2 diseases-10-00093-t002:** Household prevalence of schistosomiasis at the province and village levels, determined by real-time PCR.

		Total Number	Number Positive	Percentage Positive	95% CI *
Province	Hunan	1099	141	12.8	10.8–14.8
	Jiangxi	1250	33	2.6	1.8–3.5
	Combined	2349	174	7.4	6.3–7.4
Hunan	Changjiang	98	7	7.1	2.0–12.
	Dongtinghong	175	8	4.6	1.4–7.7
	Ganzhou	197	23	11.7	7.2–16.2
	Longwang	104	29	27.9	19.1–36.6
	Xiangjiang	61	24	39.3	26.7–52.0
	Xincheng	173	23	13.3	8.2–18.4
	Xinggang	118	6	5.1	1.1–9.1
	Zhangshu	173	21	12.1	7.2–17.1
Jiangxi	Biaoen	146	8	5.5	1.7–9.2
	Caohui	118	3	2.5	0–5.4
	Dingshan	132	0	0.0	na
	Dongfang	183	7	3.8	1.0–6.6
	Fuqian	156	4	2.6	0–5.1
	Huanggin	172	2	1.2	0–2.8
	Huangjia	222	6	2.7	0.6–4.9
	Xinhua	121	3	2.5	0–5.3

* 95% confidence interval.

**Table 3 diseases-10-00093-t003:** The association of covariates and schistosome infection in Hunan and Jiangxi Provinces, derived from a logistic regression model.

Variable	Hunan Province	Jiangxi Province
	Regression Coefficients (95% CrI)	Regression Coefficients (95% CrI)
Distance to waterbody	**−2.367 (−4.217, −0.697)**	**−5.471 (−11.332, −0.507)**
Distance to health facilities	**−0.424 (−0.637, −0.236)**	−0.056 (−0.415, 0.269)
Altitude	**−4.947 (−7.228, −2.727)**	5.537 (−0.996, 10.014)
Temperature	−0.127 (−0.448, 0.198)	**2.664 (0.964, 4.360)**
Intercept	−4.789 (−6.077, −3.667)	−6.405 (−10.387, −3.203)

Bold shows ‘statistically significant’ results within a Bayesian framework (no zero within the 95% CrI).

**Table 4 diseases-10-00093-t004:** Regression coefficient mean and 95% credible intervals (CrI) of covariates included in a Bayesian spatial model with Binomial response for the prevalence of schistosomiasis.

Variable	Hunan Province	Jiangxi Province
	Regression Coefficients (95% CrI)	Regression Coefficients (95% CrI)
Distance to waterbody	**−1.158 (−2.104, −0.116)**	2.821 (−3.508, 8.722)
Distance to health facilities	−0.079 (−0.301, 0.101)	0.291 (−0.011, 0.621)
Altitude	2.712 (−7.542, 11.409)	15.888 (−12.109, 41.995)
Temperature	0.320 (−0.477, 0.901)	**−4.359 (−9.641, −0.055)**
Intercept	−1.597 (−8.544, 3.860)	4.383 (−9.716, 17.014)

Bold shows ‘statistically significant’ results within a Bayesian framework (no zero within the 95% CrI).

## Data Availability

All data can be found either in this manuscript or in Gordon, C. A., G. M. Williams, D. J. Gray, A. C. Clements, X.-N. Zhou, Y. Li, J. Utzinger, J. Kurscheid, S. Forsyth and K. A. Alene (2022). “Schistosomiasis in the People’s Republic of China–down but not out.” Parasitology 149(2): 218–233.

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
