# Peer review of "Spatial Analysis of Schistosomiasis in Hunan and Jiangxi Provinces in the People’s Republic of China"

_diseases, 2022, doi:10.3390/diseases10040093_

Round 1

Reviewer 1 Report

General comment

The paper presents a spatial analysis that aimed to determine risk factors associated with Schistosoma japonicum infection in Hunan and Jiangxi provinces in the People’s Republic of China. The paper is well developed, following a robust methodological approach. The paper presents a diverse set of findings that are nicely presented, including tables and figures. In the discussion chapter, environmental and socio-economic differences between the two provinces could be more systematically compared, which may then lead to more specific/novel conclusions.

Specific comments

Title:

The title accurately reflects what the paper is about.

Abstract:

The abstract is well developed and features a clear structure. Since the paper presents survey data, it would be good to present that sample size so that the reader has a feeling of the robustness of the paper. In conjunction with this, it is worth to consider adding the confidence intervals for the prevalence rates presented.

Key words:

Suggesting to feature key words in alphabetical order. Schistosoma japonica needs to be written in italics.

Introduction:

The introduction is well developed. Consider adding a few additional sentences explaining why the research undertaken is relevant in Hunan and Jiangxi provinces. What is the knowledge gap the paper aims to address?

Last paragraph of the introduction: cannot judge whether this is needed. Superscrip needed for 70”th”. Opening quotation mark needed (or closing quotatoin mark to be removed) in last sentence.

Materials and Methods:

The methods chapter is well developed. Specific observations:

= Study design and settings: suggesting to indicate when the survey was conducted. It is mentioned that the data presented was part of a longitudinal cohort study. This leads me to the question why only the data from the baseline data is presented and not the longitudinal data (which should be available by now) à On a side note: according to the abstract, the data was generated in 2016. Any possibility to justify why this data is published 6 years after the baseline survey was done?

= Line 102: space needed before intext reference.

= Check whether all abbreviations have been properly introduced (e.g. qPCR, DNA)

= It would be nice to have a map showing the study setting (à discovered Figure 1 only at the end of the manuscript à recommending to either move Figure 1 to the methods section or at least provide reference to Figure 1 in the methods)

= Line 194: check font size of in text reference

Results:

The results are comprehensive and well developed. Specific observations:

= Can the sex ratio of participants be included (also in table 1)?

= Consider providing the confidence interval for prevalence rates.

Discussion:

The main findings are discussed in detail and the description of the limitations of the study seems comprehensive. Specific observations:

= Line 282: “Recent estimates” does not seem appropriate to me as references date back to 2013-2014. Are there no more recent estimates available?

= While the authors elaborate in detail on potential factors that explain why distance from water is associated with S. japonicum transmission, the differences between Hunan and Jiangxi province in terms of the factors described are not specified. This renders it difficult to identify the main factors that explain why distance from water was significantly negatively associated with S. japonicum in Hunan but not Jiangxi province. The authors are invited to more systematically reflect on environmental and socio-economic differences between the two provinces as part of the discussion chapter, which might help to come to a more specific conclusion

Conclusions:

In my opinion, the conclusion is too much of a repetition of what is already well-established knowledge. Can the authors come up with more specific conclusions (see also my last comment on the discussion chapter), which could then also be featured in the abstract?

Author Response

Reviewer 1

General comment

The paper presents a spatial analysis that aimed to determine risk factors associated with Schistosoma japonicum infection in Hunan and Jiangxi provinces in the People’s Republic of China. The paper is well developed, following a robust methodological approach. The paper presents a diverse set of findings that are nicely presented, including tables and figures. In the discussion chapter, environmental and socio-economic differences between the two provinces could be more systematically compared, which may then lead to more specific/novel conclusions.

Response: We thank Reviewer #1 for the overall positive appraisal of our research. As requested, we now discuss differences in environmental and socio-economic features of the two provinces in some more detail (see revised manuscript, lines 363-369).

Specific comments

Title:

The title accurately reflects what the paper is about.

Response: Thanks!

Abstract:

The abstract is well developed and features a clear structure. Since the paper presents survey data, it would be good to present that sample size so that the reader has a feeling of the robustness of the paper. In conjunction with this, it is worth to consider adding the confidence intervals for the prevalence rates presented.

Response: Due to the tight word limit in the Abstract we could not include additional information. However, we have provided the detailed information, including sample size and confidence intervals, in the main Results sections of the revised version of the manuscript (see revised manuscript, Tables 1 and 2).

Key words:

Suggesting to feature key words in alphabetical order. Schistosoma japonica needs to be written in italics.

Response: Keywords have now been ordered alphabetically and species names italicized (see revised manuscript, lines 39-40).

Introduction:

The introduction is well developed. Consider adding a few additional sentences explaining why the research undertaken is relevant in Hunan and Jiangxi provinces. What is the knowledge gap the paper aims to address?

Response: As requested, additional content has been added at the end of the Introduction (see revised manuscript, lines 82-86).

Last paragraph of the introduction: cannot judge whether this is needed. Superscrip needed for 70”th”. Opening quotation mark needed (or closing quotatoin mark to be removed) in last sentence.

Response: The quotation mark has been removed

Materials and Methods:

The methods chapter is well developed. Specific observations:

= Study design and settings: suggesting to indicate when the survey was conducted. It is mentioned that the data presented was part of a longitudinal cohort study. This leads me to the question why only the data from the baseline data is presented and not the longitudinal data (which should be available by now) à On a side note: according to the abstract, the data was generated in 2016. Any possibility to justify why this data is published 6 years after the baseline survey was done?

Response: The date of data collection has been added. There have been a number of factors that have impeded speed of data production and analysis; major interruptions due to the ongoing COVID-19 pandemic have been especially problematical. However, as our piece is already quite long and many studies experienced delays in face of the major pandemic, no further action has been taken.

= Line 102: space needed before intext reference.

Response: Space added

= Check whether all abbreviations have been properly introduced (e.g. qPCR, DNA)

Response: We carefully checked the entire piece and introduced abbreviations upon first use. For instance, we now added “qPCR” (see revised manuscript, line 98).

= It would be nice to have a map showing the study setting (à discovered Figure 1 only at the end of the manuscript à recommending to either move Figure 1 to the methods section or at least provide reference to Figure 1 in the methods)

Response: Reference to Figure 1 is now provided in the Methods section (see revised manuscript, line 95).

= Line 194: check font size of in text reference

Response: We thank Reviewer #1 for checking our piece so carefully. The font size has been changed

Results

The results are comprehensive and well developed. Specific observations:

= Can the sex ratio of participants be included (also in table 1)?

Response: We have now added the sex ratio of the participants (see revised manuscript, table 1)

= Consider providing the confidence interval for prevalence rates.

Response: The confidence interval for the prevalence rates have been added (see revised manuscript, Table 2).

Discussion:

The main findings are discussed in detail and the description of the limitations of the study seems comprehensive. Specific observations:

= Line 282: “Recent estimates” does not seem appropriate to me as references date back to 2013-2014. Are there no more recent estimates available?

Response: We have updated with more recent numbers from Hunan and Jiangxi provinces; i.e. 2018 (see revised manuscript, lines 293-294).

= While the authors elaborate in detail on potential factors that explain why distance from water is associated with S. japonicum transmission, the differences between Hunan and Jiangxi province in terms of the factors described are not specified. This renders it difficult to identify the main factors that explain why distance from water was significantly negatively associated with S. japonicum in Hunan but not Jiangxi province. The authors are invited to more systematically reflect on environmental and socio-economic differences between the two provinces as part of the discussion chapter, which might help to come to a more specific conclusion

Response: We have now made this clear in the revised version of the manuscript that distance to waterbody was significantly associated with S. japonicum in Hunan province but not in Jiangxi province. The difference in environmental and socio-demographic variables between the two provinces might influence the association of distance to waterbody and S. japonicum. However, further nationwide studies using genomic and survey data might be required to better understand why distance to waterbody was negatively associated with distribution of schistosomiasis in Hunan province but not in Jiangxi province.

Additionally, the distance from water in villages from both provinces was not significantly different, which means the variable may not be a great predictor in this instance as the observations are quite homogenous between all villages.

In our previous publication (Gordon et al., 2022 [PMID: 35234601]), we noted that the two populations in Jiangxi villages and Hunan villages, were quite homogenous with similar levels of education and the majority in occupation as farmers and/or fishermen, and similar socio-economic levels, as determined by calculating a wealth index. Transmission dynamic between the two regions (Dongting Lake and Poyang Lake) are also similar. Some of this information has been included in the Discussion (see revised manuscript, lines 320, 385-391).

Conclusions:

In my opinion, the conclusion is too much of a repetition of what is already well-established knowledge. Can the authors come up with more specific conclusions (see also my last comment on the discussion chapter), which could then also be featured in the abstract?

Response: As suggested by Reviewer #1, we have now revised the Conclusions, as follow: “In this study, the spatial distribution of schistosomiasis in Hunan and Jiangxi varied substantially at the provincial, village, and local levels, and was significantly associated with distance to nearest waterbody and climatic factors. We produced maps showing the spatial distribution of schistosomiasis in Hunan and Jiangxi provinces. These maps may help the local governments to design strategies and target interventions for the schistosomiasis control and elimination programmes. Interventions should ideally be targeted to individuals who live in close proximity to open water sources, as they are at highest risk of infection and the negative implications of schistosomiasis japonica. However, further nationwide studies using genomic and survey data might be required to better understand why distance to waterbody was negatively associated with distribution of schistosomiasis in Hunan province but not in Jiangxi province.”

Reviewer 2 Report

The article “Spatial analysis of schistosomiasis in Hunan and Jiangxi provinces in the People’s Republic of China” describes results from spatial analysis to determine risk factors that may impact on Schistosoma japonicum infection and predict risk in Hunan and Jiangxi provinces in the People’s Republic of China.
In my opinion the manuscript is well written and interesting for epidemiologist and general readers.

I have only one small issue for the authors: could you include in your paper name of extraction kit and briefly description of qPCR?

Apart from this one minor suggestion, I have nothing more to add.

Author Response

The article “Spatial analysis of schistosomiasis in Hunan and Jiangxi provinces in the People’s Republic of China” describes results from spatial analysis to determine risk factors that may impact on Schistosoma japonicum infection and predict risk in Hunan and Jiangxi provinces in the People’s Republic of China.
In my opinion the manuscript is well written and interesting for epidemiologist and general readers.

I have only one small issue for the authors: could you include in your paper name of extraction kit and briefly description of qPCR?

Apart from this one minor suggestion, I have nothing more to add.

Response: We thank Reviewer #2 for the generous comments. In the meantime, we added the requested details for the extraction kit (see revised manuscript, line 103). Please note that full qPCR details are provided in reference #6, and hence, no further action has been taken.

Reviewer 3 Report

The manuscript entitled “Spatial analysis of schistosomiasis in Hunan and Jiangxi provinces in the People’s Republic of China”, submitted by Alene and colleagues, predicts the prevalence of Schistosoma japonicum in the provinces of Hunan and Jiangxi using geological and climatic variables and prevalence data from a previous study. From their Bayesian spatial model, they found a negative effect from distance of waterbody in Hunan province and a negative effect from temperature in the Jiangxi province. Overall, the manuscript is well-written and the statistical analyses appear to be carefully constructed. The objective of the study is pretty straightforward and, though the predictive power of the models is weak outside of the sampling areas, due to the limitations of the data from the previous study, I think that this study is worthy, since it can still provide valuable information for knowledge users and targeted elimination programmes. Please address the minor comments and concerns that I have listed below.

Line 26: Please change for “that may impact Schistosoma japonica infection”

Line 44: Change “,” instead of “.”

Line 57: Refer to the study as Author (Year) because it is not the exact same list of authors as in the current study.

Line 80: Write the species name correctly (capitalise and italicise).

Line 87: Remove quotation mark.

Line 101: Rewrite for “Garmin Handheld GPS (Kansas, USA)”

Line 102: Capitalise letters in “Global Lakes and Wetlands Database [19]”

Line 110: Remove this sentence, as it is already written at the beginning of this paragraph.

Line 119: Cite the studies and resources you are referring to here.

Line 121: It is typically written in one word “waterbody”, so make this consistent throughout the text. Also, do you specifically mean still water (lakes, ponds) or also include moving water (rivers, streams)? Please specify this here.

Lines 124-126: Rewrite the text in the parentheses so you do not start a new sentence in the parentheses, which is grammatically incorrect.

Line 133: If you are using a spatial resolution of 0.1 X 0.1 km, then how do you get 0.5 km2 grid cells? Shouldn’t it be 0.01 km2 grid cells? Also, I do not understand why you used these different buffer zones, please provide an explanation in the text as to how this differs from your grid cells.

Line 182: Put the -4 in superscript.

Line 189: Please include the version of R used for the analyses.

Line 202: Perhaps provide the median distance instead, which could be more informative to the reader?

Line 203: Change “measures” to “measure”

Lines 242-247: No need to repeat the regression coefficients and credible intervals since you already present them in Table 3.

Lines 255-268: Again, there is no need to repeat the values if they are already in a table.

Line 263: Should be “presents” instead of “present”

Line 276: Should be “over the majority”

Lines 288-289: What are you basing the negative trend in both provinces on, the Bayesian spatial model? Because the regression coefficient for Jiangxi province is positive, not negative. Please clarify what you are basing this on.

Line 335: That is not true, only Table 3 reports a positive correlation with temperature in Jiangxi province, please correct this.

Lines 355 and 357: Do not capitalise schistosomiasis.

Line 368: Write the species name correctly (capitalise and italicise).

Figure 2: Please provide the size of grid cells in the figure caption.

Author Response

The manuscript entitled “Spatial analysis of schistosomiasis in Hunan and Jiangxi provinces in the People’s Republic of China”, submitted by Alene and colleagues, predicts the prevalence of Schistosoma japonicum in the provinces of Hunan and Jiangxi using geological and climatic variables and prevalence data from a previous study. From their Bayesian spatial model, they found a negative effect from distance of waterbody in Hunan province and a negative effect from temperature in the Jiangxi province. Overall, the manuscript is well-written and the statistical analyses appear to be carefully constructed. The objective of the study is pretty straightforward and, though the predictive power of the models is weak outside of the sampling areas, due to the limitations of the data from the previous study, I think that this study is worthy, since it can still provide valuable information for knowledge users and targeted elimination programmes. Please address the minor comments and concerns that I have listed below.

  1. Line 26: Please change for “that may impact Schistosoma japonicainfection”
  2. Line 44: Change “,” instead of “.”
  3. Line 57: Refer to the study as Author (Year) because it is not the exact same list of authors as in the current study.
  4. Line 80: Write the species name correctly (capitalise and italicise).

Response: This is referring to the disease rather than the species, thus no further action has been taken.

  1. Line 87: Remove quotation mark.
  2. Line 101: Rewrite for “Garmin Handheld GPS (Kansas, USA)”
  3. Line 102: Capitalise letters in “Global Lakes and Wetlands Database [19]”
  4. Line 110: Remove this sentence, as it is already written at the beginning of this paragraph.

Response: Comments 1-8 all completed.

  1. Line 119: Cite the studies and resources you are referring to here.

Response: References added

  1. Line 121: It is typically written in one word “waterbody”, so make this consistent throughout the text. Also, do you specifically mean still water (lakes, ponds) or also include moving water (rivers, streams)? Please specify this here.

Response: Lakes, ponds, rivers and streams all included. This has been added to the manuscript.

  1. Lines 124-126: Rewrite the text in the parentheses so you do not start a new sentence in the parentheses, which is grammatically incorrect.

Response: This has been changed to a proper sentence, although it was supposed to just be a description of a term used within the sentence – natural village.

  1. Line 133: If you are using a spatial resolution of 0.1 X 0.1 km, then how do you get 0.5 km2grid cells? Shouldn’t it be 0.01 km2 grid cells? Also, I do not understand why you used these different buffer zones, please provide an explanation in the text as to how this differs from your grid cells.

Response: This has been updated to ‘centroid of each natural village with 500 m radii’ (see revised manuscript, line 142).

  1. Line 182: Put the -4 in superscript.

Response: This has been modified as requested

  1. Line 189: Please include the version of R used for the analyses.

Response: The respective version of R has been added (see revised manuscript, line 157).

Line 202: Perhaps provide the median distance instead, which could be more informative to the reader?

Response: We have now provided the median and interquartile ranges in the revised version of the manuscript (see revised manuscript, line 215).

Line 203: Change “measures” to “measure”

Response: Done (see revised manuscript, line xx).

Lines 242-247: No need to repeat the regression coefficients and credible intervals since you already present them in Table 3.

Response: Done

Lines 255-268: Again, there is no need to repeat the values if they are already in a table.

Response: Done

Line 263: Should be “presents” instead of “present”

Response: Done

Line 276: Should be “over the majority”

Response: Done

Lines 288-289: What are you basing the negative trend in both provinces on, the Bayesian spatial model? Because the regression coefficient for Jiangxi province is positive, not negative. Please clarify what you are basing this on.

Response: Corrected to indicate only referring to Hunan province in this statement

Line 335: That is not true, only Table 3 reports a positive correlation with temperature in Jiangxi province, please correct this.

Response: This issue has been corrected

Lines 355 and 357: Do not capitalise schistosomiasis.

Response: This has been corrected

Line 368: Write the species name correctly (capitalise and italicise).

Response: Here, we were referring to the disease rather than the species here. Hence, no further action taken.

Figure 2: Please provide the size of grid cells in the figure caption.

Response: Done (see revised manuscript, Figure 2, caption).

In addition to comments put forth by the three reviewers, Figure 2 has been updated to use grid sizes of approximately 0.1 km2 and restricted borders to closer match with where study data were collected rather than applying to whole province